# Deciphering the Genomic Landscape and Pharmacological Profile of Uncommon Entities of Adult Rhabdomyosarcomas

**DOI:** 10.3390/ijms222111564

**Published:** 2021-10-26

**Authors:** Alessandro De Vita, Silvia Vanni, Valentina Fausti, Claudia Cocchi, Federica Recine, Giacomo Miserocchi, Chiara Liverani, Chiara Spadazzi, Massimo Bassi, Manlio Gessaroli, Angelo Campobassi, Giovanni De Luca, Federica Pieri, Anna Farnedi, Eugenia Franchini, Anna Ferrari, Chiara Domizio, Enrico Cavagna, Lorena Gurrieri, Alberto Bongiovanni, Nada Riva, Sebastiano Calpona, Giandomenico Di Menna, Silvia Angela Debonis, Toni Ibrahim, Laura Mercatali

**Affiliations:** 1Osteoncology and Rare Tumors Center, IRCCS Istituto Romagnolo per lo Studio dei Tumori (IRST) “Dino Amadori”, 47014 Meldola, Italy; silvia.vanni@irst.emr.it (S.V.); valentina.fausti@irst.emr.it (V.F.); federica.recine@irst.emr.it (F.R.); giacomo.miserocchi@irst.emr.it (G.M.); chiara.liverani@irst.emr.it (C.L.); chiara.spadazzi@irst.emr.it (C.S.); lorena.gurrieri@irst.emr.it (L.G.); alberto.bongiovanni@irst.emr.it (A.B.); nada.riva@irst.emr.it (N.R.); sebastiano.calpona@irst.emr.it (S.C.); giandomenico.dimenna@irst.emr.it (G.D.M.); silviaangela.debonis@irst.emr.it (S.A.D.); toni.ibrahim@irst.emr.it (T.I.); laura.mercatali@irst.emr.it (L.M.); 2Medical Oncology Unit, Azienda Ospedaliera “San Giovanni Addolorata”, 00184 Roma, Italy; 3Oral and Maxillofacial Surgery Unit, “Maurizio Bufalini” Hospital, 47521 Cesena, Italy; massimo.bassi@auslromagna.it (M.B.); manlio.gessaroli@auslromagna.it (M.G.); angelo.campobassi@auslromagna.it (A.C.); 4Pathology Unit, “Maurizio Bufalini” Hospital, 47521 Cesena, Italy; giovanni.deluca@auslromagna.it; 5Pathology Unit, “Morgagni-Pierantoni” Hospital, 47121 Forlì, Italy; federica.pieri@auslromagna.it (F.P.); anna.farnedi@auslromagna.it (A.F.); 6Biosciences Laboratory, IRCCS Istituto Romagnolo per lo Studio dei Tumori (IRST) “Dino Amadori”, 47014 Meldola, Italy; eugenia.franchini@irst.emr.it (E.F.); anna.ferrari@irst.emr.it (A.F.); chiara.domizio@irst.emr.it (C.D.); 7Department of Diagnostic Imaging, Azienda Unità Sanitaria Locale della Romagna (AUSL Romagna), 47923 Rimini, Italy; enrico.cavagna@auslromagna.it

**Keywords:** adult rhabdomyosarcoma, primary culture, gene expression profiling, chemotherapy, 3D collagen-based scaffold culture systems

## Abstract

Adult rhabdomyosarcoma (RMS) represents an uncommon entity with an incidence of less than 3% of all soft tissue sarcomas (STS). Consequently, the natural history and the clinical management of this disease are infrequently reported. In order to fill this gap, we investigated the molecular biology of an adult RMS case series. The expression of epithelial mesenchymal transition-related gene and chemoresistance-related gene panels were evaluated. Moreover, taking advantage of our STS translational model combining patient-derived primary culture and 3D-scaffold, the pharmacological profile of an adult head and neck sclerosing RMS was assessed. Furthermore, NGS, microsatellite instability, and in silico analyses were carried out. RT-PCR identified the upregulation of *CDH1*, *SLUG*, *MMP9*, *RAB22a*, *S100P,* and *LAPTM4b*, representing promising biomarkers for this disease. Pharmacological profiling showed the highest sensitivity with anthracycline-based regimen in both 2D and 3D culture systems. NGS analysis detected *RAB3IP-HMGA2* in frame gene rearrangement and *FGFR4* mutation; microsatellite instability analysis did not detect any alteration. In silico analysis confirmed the mutation of *FGFR4* as a promising marker for poor prognosis and a potential therapeutic target. We report for the first time the molecular and pharmacological characterization of rare entities of adult head and neck and posterior trunk RMS. These preliminary data could shed light on this poorly understood disease.

## 1. Introduction

Rhabdomyosarcoma (RMS) represents one of the most frequent soft tissue sarcomas (STS) of children but is very uncommon in adult STS, accounting for less than 3%. Due to the rarity of adult RMS, limited information about biological features, prognostic factors, and response to chemotherapy are currently available. For this reason, the current management of adult RMS is often based on studies done in the pediatric setting. However, adult RMS differs from the childhood counterpart in terms of natural history, clinical behavior, and outcomes. In particular, adult RMS affects different body sites with a predilection for extremities [1], while children RMS shows a propensity for head and neck (35%), genitourinary tract (22%), and extremities (18%) [2,3]. In adult RMS, the most common site for metastasis is lungs (50%), followed by bone marrow, lymph nodes, and, less frequently, the brain [4,5]. For children RMS, the most reported sites for metastasis are lungs (47%), which represent the only dissemination site for 18% of metastatic patients, followed by bone marrow (38%), bone (34%), and distant lymph nodes (26%) [6].

Morphologically, RMS can be classified into embryonal, alveolar, pleomorphic, and spindle cell/sclerosing rhabdomyosarcoma [7,8,9]. Embryonal (49%) RMS represents the most common subtype for the entire population, followed by alveolar one (30%). The most common subtypes for adult RMS include pleomorphic and spindle/sclerosing variants, although embryonal and alveolar RMS have also been reported [10]. Otherwise, the most predominant entities of children RMS are embryonal and alveolar subtypes [11].

Histopathologically, embryonal RMS is characterized by sheets of rounded spindle cells with eosinophilic rhabdomyoblasts, while alveolar RMS consists of undifferentiated uniform round cells in an alveolar pattern or in diffuse sheets with sporadic characteristic multinucleated giant cells. Pleomorphic RMS instead exhibits sheets and fascicles of highly pleomorphic cells with brightly eosinophilic cytoplasm. Finally, sclerosing RMS consists of dense hyaline matrix-containing lobules and infiltrating cords of small round malignant cells without calcification.

Diagnosis is based on morphological and immunohistochemical findings, with positivity for MyoD1, desmin, and myosin [12,13].

The 5-year overall survival (OS) in adult RMS is poor, ranging from about 20–40% [1,14], while the OS of the children variant is about 77–87% [3,15].

Current treatment strategies may include surgery, radiotherapy, and chemotherapy.

For localized disease, surgery remains the main option, however, tumor shrinkage may be achieved with anthracycline-based neoadjuvant chemotherapy. Chemosensitivity is strongly related to the histological subtype, ranging from a good chemosensitivity for ERMS (embryonal RMS) and ARMS (alveolar RMS) to a poor sensitivity to systemic treatment in more aggressive histological entities such as pleomorphic RMS [16]. Post-surgery radiotherapy, regardless of the margin status, demonstrated to improve outcome in terms of local and distant recurrence as well as overall survival (OS) compared to patients with adult RMS that did not receive adjuvant radiotherapy [17]. Treatment of metastatic disease remains controversial. Pediatric protocol of polychemotherapy regimen (mainly based on the combination of ifosfamide, vincristine, and actinomycin ± maintenance chemotherapy with metronomic cyclophosphamide) has shown to provide little improvement in terms of ORR compared to adult-standard STS regimens (single-agent doxorubicin or ifosfamide-epirubicin). However, this protocol is not always suitable for adults due to its toxicity [18,19]. Given the rarity and complexity of this disease, multidisciplinary management in referral centers is crucial to improve outcomes [20].

Until now, limited and anecdotal analyses have been performed on adult RMS in terms of gene expression profiling aimed to dissect the biology of these uncommon lesions. In particular, recent works have addressed the role of TFCP2 fusions and ALK overexpression as predictors of poor prognosis in adult RMS with mixed epithelioid and spindle morphology [21]. Moreover, spindle cell and sclerosing RMS harboring either homozygous or heterozygous MYOD1 (p. L122R) exon 1 mutations have been identified as aggressive RMS subset compared to non-mutated histotypes [22].

For the above reasons, there is a pressing need to identify novel molecular markers for a better patient stratification in order to guide clinicians in the therapy selection. Finally, a more in-depth investigation on the role of chemotherapy is needed in order to identify which type of patient could be eligible for current treatments and which not.

This exploratory study tries to shed light on this poorly investigated disease, providing new insight about the molecular landscape and pharmacological profile of adult head and neck and posterior trunk RMS.

## 2. Results

### 2.1. Patient Clinicopathological Characteristics

RMS1 was a 74 year-old male patient with a diagnosis of head and neck rhabdomyosarcoma. In February 2019, the patient underwent a complex surgery consisting of a resection of the local disease with right lateral cervical lymphadenectomy, removal of parotid, and mandibular demolition with reconstruction by right peroneal flap performed by a specialized team of maxillofacial surgeons (Appendix A). The histology reported the diagnosis of high grade sclerosing rhabdomyosarcoma of the masseter parotid region of about 7 cm with positive surgical margins (bone), 60% Ki-67 positivity and one lymph node as site of micro-metastasis. Subsequently, the patient received an adjuvant IMRT radiotherapy treatment, 60 Gy divided in 30 fractions, that was completed in May 2019. In July 2019, the patient was referred to our Center with the appearance in the MRI of multiple nodules in the sternocleidomastoid muscle and subcutaneous carcinosis (Figure 1a). The biopsy of the nodules confirmed the diagnosis of rhabdomyosarcoma. The CT scan showed a laterocervical lymphoadenopathy of 75 mm × 65 mm and several pulmonary and liver metastases. The multidisciplinary board suggested chemotherapy with doxorubicin 75 mg/m^2^ day 1 every 3 weeks. The radiological evaluation after four cycles of treatment showed progression of the disease (Appendix A). Due to the persistence of a good performance status (PS ECOG 1), in December 2019, the patient received a second line of treatment with Gemcitabine 1000 mg/m^2^ on day 1, 8, and 15 every 4 weeks. After three cycles of treatment, the patient achieved a stable disease according to RECIST 1.1 criteria and reported a clinical benefit. Treatment was discontinued in June 2020 after clinical evidence of progressive disease and the patient died on 6 July 2020. 

RMS2 was a 72 year-old female patient with a diagnosis of head and neck rhabdomyosarcoma. Medical history reported bladder prolapse operated in 2007, uterine polypectomy in 2010, right hip arthroplasty surgery in 2015, video laparoscopic cholecystectomy in 2015, and hiatal hernia and gastroesophageal reflux, as well as findings of gastric ulcer treated with proton pump inhibitors and kidney colic from uric acid stones diverticulosis. In November 2020, the patient underwent hospitalization in an infectious disease unit for SARS-CoV-2 pneumonia (COVID-19). In December 2020, the patient complained about diplopia and a subsequent eyes examination reported ophthalmodynia, eyelid edema, and right painful exophthalmos. After the subsequent MRI examination (Figure 1b) and maxillofacial consultation, a biopsy was performed in January 2021. The diagnosis was of alveolar rhabdomyosarcoma of right ethmoid-orbital region.

RMS3 was a 76-year-old male patient with a diagnosis of latissimus dorsi muscle rhabdomyosarcoma. Medical history reported arterial hypertension, previous episodes of heart failure, pulmonary sarcoidosis, and renal angiomyolipomas in follow-up. At the end of February 2021, a rapidly worsening nodule on the right lateral chest wall appeared. Ultrasound analysis performed in March 2021 showed appreciable swelling corresponding to a gross substantially isoechoic solid lesion, localized in the thickness of the muscle plane, with clear margins and size of about 5.1 cm × 2.8 cm. The lesion was devoid of vascular signal and caused compression of the surrounding planes. The subsequent PET analysis performed in April 2021 showed a voluminous intensely hypermetabolic mass with a diameter of 7.7 cm × 5 cm and maximum SUV equal to 17, located at the level of the right thoracic lateral wall. No other lesions were reported. In June 2021, the patient underwent excision of right dorsal neoformation (previously performed biopsy was positive for spindle cell type rhabdomyosarcoma) and the diagnosis was pleomorphic rhabdomyosarcoma. Surgery achieved an R0 resection, so no adjuvant treatment was indicated.

Clinical pathologic characteristics of RMS patients enrolled in the study are provided in Table 1.

### 2.2. Diagnosis of RMS Case Series

Macroscopic evaluation of surgically-resected tumor tissue of RMS1 patient revealed a soft tissue mass of the palotid loggia measuring 10 cm × 9 cm × 3 cm, with attached axillary cable in block measuring 15 cm × 8 cm. When cut, there was a nodular lesion of 7 cm in diameter with a soft-elastic consistency and yellow-gray color in correspondence to the palotid loggia. Hematoxylin and eosin-stained tumor tissue (Figure 2) reviewed by an experienced sarcoma pathologist revealed a densely cellularized malignant mesenchymal neoplasia with infiltrative and multinodular growth pattern. The lesion was made up of round cells, with hyperchromic and pleomorphic nuclei, arranged in cords, trabeculae, and microalveolar structures buried by dense sclerotial stroma. Focal tumor necrosis was present (<50%) and focal vascular invasion was observed as well. Numerous medium and large caliber nerve bundles were recognizable within the neoplasm. Bone tissue was fiercely infiltrated by neoplasm. Mitotic index was 20 MF × 10 HPF. Cell proliferation index, evaluated on immunohistochemical reaction with Ki-67/Mib1, was equal to 60%. Immunohistochemical investigations revealed that neoplastic cells were diffusely positive for vimentin, MyoD1 and CD56, and focally positive for desmin, myoglobin, myogenin, actin ML, CD34, and AE1-AE3 cytokeratin (Figure 2). Tumor cells were negative for S100, STAT6, and P63. Moreover, a widespread nuclear positivity for MDM2 was detected (Figure 2). One metastasis measuring 2 mm, out of 20 isolated lymph nodes, was observed. No evidence of metastasis in glandular parenchyma, peri-lymph node soft tissue extension, and fibroadipose tissue was reported. The diagnosis was sclerosing rhabdomyosarcoma grade 3 FNCLCC with positive surgical margin.

Incisional sampling biopsy of right orbital ethmoid neoformation of RMS2 patient revealed a grayish and translucent nodular fragment measuring 1.2 cm × 0.8 cm. Hematoxylin and eosin-stained tumor tissue (Figure 3) reviewed by an experienced sarcoma pathologist revealed fragments of round cells malignant mesenchymal neoplasm, densely cellulated, with aspects of rhabdoids and alveolar growth patterns. Mitotic index was 10 MF × 10 HPF. Cell proliferation index, evaluated on immunohistochemical reaction with Ki-67/Mib1, was equal to 50%. Immunohistochemical investigations revealed that neoplastic cells were intensely and diffusely positive for Desmin, Myogenin, and MYOD1 (Figure 3), and weakly positive for synaptophysin and negative for CD99, Cytokeratin AE1-AE3, GFAP, CD45, S100, and MDM2 (Figure 3). The diagnosis was alveolar rhabdomyosarcoma.

Macroscopic evaluation of surgically-resected tumor tissue of RMS3 patient revealed a piece of 600 g of 15 cm × 11.5 cm × 7.5 cm consisting of striated muscle tissue bordered by fibroadipose tissue, comprising a whitish nodule with clear margins measuring 8 cm × 7.5 cm × 6 cm. A cutaneous lozenge of 7 cm × 2 cm was also present. Hematoxylin and eosin-stained tumor tissue (Figure 4) reviewed by an experienced sarcoma pathologist revealed a neoplasm mostly made up of pleomorphic cellular elements of variable size, with rather uniform spindle cell areas showing a fasciculate pattern. Numerous mitotic figures were observed, some of which were atypical ones. The lesion showed defined margins and appeared completely excised. No evidence of infiltration of the muscular and fibroadipose tissue was reported while skin showed dermal scar. Immunohistochemical investigations revealed that cells were positive for desmin; focally positive for MYoD1 and myogenin; focally and weakly positive for smooth muscle actin; and negative for CD34, cytokeratin AE1-AE3, and S100 (Figure 4). The index cell proliferation, evaluated with Ki67, was equal to 25%. The diagnosis was pleomorphic rhabdomyosarcoma.

### 2.3. Tumor Tissue Gene Expression Analyses

In order to gain better insight into the genomic landscape of these uncommon entities, panels of EMT- and chemoresistance-related genes were investigated (Figure 5). In particular, in RMS1 the expression of *VIM*, an EMT gene and a master regulator of cell architecture, was 2.88-fold higher with respect to control tissue. *CDH1*, an epithelial marker, was significantly upregulated (158.46 fold-higher, *p* < 0.05) with respect to control. *TGFb*, a major driver of EMT and a strong promoter of several tumor-associated pathways, was upregulated (2.80 fold-higher, *p* < 0.05) as well as *SNAIL1* (3.07 fold-higher *p* < 0.05). The expression levels of both matrix-modifying enzymes *MMP2* and *MMP9* were 1.31 (*p* < 0.05) and 11,890.70 fold-higher, respectively. Moreover, the expression level of *SLUG*, an inducer of EMT, was upregulated with respect to control (5.87 fold-higher, *p* < 0.05). Furthermore, we observed the upregulation of some chemoresistance-related genes. In particular, as previously mentioned [23], *LAPTM4b*, a gene which contributes to chemotherapy resistance, was upregulated (8.91 fold-higher) while the expression of *LAPTM4a* was comparable to the control. A significant upregulation of chemoresistance-associated genes *TP53I3* (2.27 fold-higher *p* < 0.05), *RAB22a* (6.08 fold-higher *p* < 0.05) and *S100p* (88.5 fold-higher *p* < 0.05) was detected. Finally, the expression of *CD109*, a gene previously shown [24] to be a promising diagnostic marker and a potential therapeutic target in myxofibrosarcoma, was 2.19 (*p* < 0.05) fold-higher than that of the control. In RMS2, the expression of *VIM, CDH1, TGFb, MMP2, MMP9,* and SLUG was significantly downregulated respect to control tissue, while *SNAIL1* was upregulated (1.23 fold-higher, with a trend of significance) with respect to control. Moreover a significant downregulation of all investigated chemoresistance-associated genes was observed. In RMS3, the expression of *VIM, CDH1, TGFb, SNAIL1, MMP9,* and SLUG was significantly downregulated while *MMP2* was upregulated (2.06 fold-higher, with a trend of significance) with respect to control tissue. Moreover, a significant downregulation of all investigated chemoresistance-associated genes was observed.

Establishment of sclerosing RMS1 patient-derived primary culture was achieved (Figure 6a). The cell line continued to proliferate after culture passages. Morphological analysis was carried out on both patient-derived isolated cells cultured in standard monolayer (Figure 6b) and within collagen-based scaffolds (Figure 6c). Hematoxylin and eosin-stained primary cells reviewed by an experienced sarcoma pathologist confirmed the establishment of RMS primary culture. Moreover, cytological analysis of MyoD1 in RMS patient-derived primary culture confirmed the expression of this marker in 70% (Figure 6d) of isolated primary cells.

### 2.4. Chemotherapy Assessment in 2D and 3D Patient-Derived Primary Culture RMS Model

To better elucidate the role of chemotherapy in adult RMS, we exposed the high-grade RMS1 primary cells, both cultured in 2D and within 3D collagen-based scaffold, to DOXO, to the combination of DOXO and DACA, to the combination of DOXO and CIS, and to ETO. Moreover, the activity of LENVA, an antiangiogenic molecule currently under clinical evaluation in different solid malignancies, was assessed.

RMS cells cultured in 2D showed a survival of 33% with DOXO, 31% with DOXO/DACA, 26% with DOXO/CIS, 75% with ETO, and 39% with LENVA (Figure 7a).

RMS cells cultured in 3D showed a survival of 43% with DOXO, 47% with DOXO/DACA, 43% with DOXO/CIS, ETO did not affect the survival, and 99% with LENVA (Figure 7b).

Cancer cell survival significantly decreased with respect to CTR in all 2D and 3D treatment conditions, excluding the ETO and LENVA conditions. Both types of cultures highlighted the non-inferiority of anthracyclines monoregimen therapy versus DOXO/DACA and DOXO/CIS; furthermore, the cytotoxic activity of anthracyclines monoregimen therapy resulted as significantly higher with respect to ETO and LENVA both in 2D and 3D (Figure 7).

Translational comparison between the observed clinical outcome and in vitro analysis of RMS patients enrolled in the study are provided in Table 2.

### 2.5. Next Generation Sequencing (NGS) Profiling and Microsatellite Instability (MSI) Status of Sclerosing RMS

In order to better characterize RMS1 and to provide further support to the genomic similarity of the patient-derived RMS primary culture with the tumor tissue, sequencing analysis and microsatellite instability status were analyzed in both samples. NGS profiling performed on both tumor tissue and RMS primary culture did not detect any alteration in the 52 genes included in the panel, neither in DNA nor in RNA. Both samples were comparable in number of target reads, mean coverage, and uniformity. Next, a more in-depth analysis using a RNA-seq 1385 gene panel detected *RAB3IP-HMGA2* gene rearrangement and a *FGFR4* mutation (data not shown) as previously reported [25].

MSI analysis showed microsatellite stability in both RMS patient tumor sample (Appendix A) and primary culture (Appendix A). Microsatellite instability status on both tumor tissue and RMS primary culture revealed an overlapping stable trend of the curves. In particular, all the eight investigated markers (*cKIT*, *MSH2*, *SLC7A8*, *STT3A*, *ZNF2*, *BIRC3*, *CASP2*, and *MAP4K3*) showed a ΔTm Melting Temperature (Tm sample − Tm positive control) ≥ −3 (unstable markers are considered for a ΔTm < −3) and thus were considered stable.

### 2.6. In Silico Analysis

In order to confirm the robustness of data observed in our case study, in silico analyses were performed on 43 children with RMS, which reported only somatic mutations [26]. In this regard, among the 32 most mutated genes, three cases reported the mutation of *FGFR4,* representing the 7% of analyzed patients (Appendix A). All three cases were alveolar RMS, while the mutated *FGFR4* patient detected by our analysis was affected by sclerosing histotype (Appendix A). Interestingly, the *FGFR4*-mutated RMS patient showed primary tumor in the abdominal wall, left orbital, and omentum (Appendix A). This data correlates with the primary tumor location of RMS1 patient, which affected the masseterine parotid region, underlying a possible predilection for head and neck and trunk region. Finally, in silico analysis showed high risk for all *FGFR4* mutated patients, providing evidence for the role of this marker in predicting poor prognosis as observed in the RMS1 patient.

## 3. Discussion

In this work we aimed to investigate the genomic background and the pharmacological profile of poorly understood entities of adult RMS. Thus, we conceived a retrospective study analyzing the gene expression profiling of three adult RMS cases. Moreover, a prospective study taking advantage of our patient-derived soft tissue sarcoma model combined with standard 2D and 3D culture systems was carried out on a sclerosing RMS.

First, to shed light on the molecular landscape of these diseases, we analyzed a panel of EMT and chemoresistance-related genes. The results clearly showed the upregulation of several makers compared to matched normal tissue (Figure 5). In particular, in RMS1, the upregulation of *VIM* in RT-PCR was consistent with the patient’s diagnosis in which the analyzed tumor specimen showed a diffuse positivity for vimentin with IHC and, more in general, with the mesenchymal origin of STS. Moreover, other EMT-related genes such as *TGF-b*, *SNAIL1*, *SLUG*, *MMP2,* and *MMP9* were found upregulated, confirming the above suggestion. Upregulation of *CDH1*, a well-known epithelial marker, was also detected. Although a loss of function or downregulation of *CDH1* has been associated with primary cancer and metastases [27], it has also been found that *CDH1* can be a trigger of tumor aggressiveness and metastases through the reversion of EMT to MET process [28,29,30,31,32]. The biological and clinical implication of this marker should be further investigated in order to understand its role in this disease.

Furthermore, we investigated *CD109* expression in order to provide support to our previous research [24,33] in which the role of this marker in diagnosis and as a therapeutic target in STS was pointed out, especially in myxofibrosarcoma. The slight upregulation of *CD109* detected in RMS1 provides support to the previously observed results and underlies the higher specificity of this marker for other STS histotypes. An upregulation of chemoresistance-related genes *LAPTM4b*, *TP53I3*, *RAB22a,* and *S100p* was also observed, underlying the refractoriness of these lesions to chemotherapy.

In RMS2, among all investigated EMT-related genes, an upregulation of *SNAIL1* was detected. This data was consistent with previous research that pointed out the role of *SNAIL* as a master regulator for alveolar RMS. The observed results corroborated previous research that identified this marker as a new promising target for alveolar RMS [34]. Moreover, the observed downregulation of chemoresistance-related genes reflects the good chemosensitivity of alveolar RMS, as reported above.

In RMS3, among all investigated EMT-related genes, an upregulation of *MMP9* was detected. In this regard, metallopeptidases have already been associated with tumor aggressiveness in many studies including STS and could represent therapeutic targets for these lesions [35]. Interestingly, while a chemoresistance profile was expected for a pleomorphic RMS, a downregulation of chemoresistance-related genes was observed.

Next, the establishment of a RMS patient-derived primary culture was achieved, as confirmed by an experienced sarcoma pathologist who assessed the presence of RMS tumor cells in H&E staining (Figure 6b,c). Moreover, these findings were further corroborated by MyoD1 immunohistochemical analysis, which showed 70% positivity in isolated tumor cells for this marker (Figure 6d). The surgical material also showed *MDM2* gene amplification, which is used for standard differential diagnosis of dedifferentiated liposarcoma [36]. However, its amplification has been already documented in some cases of sclerosing RMS [37,38].

The fidelity of our RMS model in recapitulating the genomic features of tumor samples was further confirmed by NGS profiling analysis, which resulted as wild type in both the tumor specimen and RMS patient-derived primary culture. We investigated the expression of a panel of tumor-associated markers known to be involved in different tumors. This analysis was carried out not only to confirm the establishment of RMS primary culture, but also the homogeneity between the original biological sample and primary culture over time. The results showed no alteration in any of the 52 genes of the panel. Since many of these markers are druggable and are used in daily clinical practice as predictive markers of response, these data represent an important indication of the possible therapeutic efficacy of some chemotherapeutic agents currently used for the management of STS patients. In particular, no alterations were detected in the expression of *ALK, MTOR, KIT, PDGFRA, NTRK1, NTRK2,* and *NTRK3,* which are the targets of crizotinib, everolimus, imatinib, and larotrectenib, respectively. These results are suggestive of the refractoriness of these diseases to several treatments currently used for the management of other STS histotypes (e.g., anthracyclines, trabectedin or immunotherapy [39,40]) and support the clinical observation of the limited availability of therapeutic options and poor outcome of adult RMS patients. The observed results prompted us to deepen the investigation of molecular features of RMS1 lesion through a larger RNA-seq panel. The results (data not shown) detected a *RAB3IP-HMGA2* gene rearrangement and a *FGFR4* mutation, which could provide some new insight about the biology of this uncommon entity [25].

Impaired DNA mismatch repair system (MMR) leads to microsatellite instability, a condition of genetic hypermutability, which is associated with different tumors, most commonly endometrial and gastric tumors [41,42]. Nevertheless, microsatellite instability wild type status resulting in both RMS tumor tissue (Appendix A) and patient-derived primary culture (Appendix A) corroborated the above observation of RMS model fidelity. Additional studies confirmed that tumors with high MSI instability status correlate with high levels of PD1/PDL1 expression [43,44] and represent predictive markers of response to immunotherapy [45] in several tumors including colorectal cancer [46,47], endometrial cancer [48], gastric cancer [49], follicular thyroid cancer [50], and head and neck carcinoma [51]. For the above reasons, the analysis of microsatellite instability represents an important tool in clinical practice and could be also used as an indirect method for the investigation of MMR system alterations, therefore, identifying the likelihood of patient responsiveness to immunotherapy. As a result, this data provides evidence of the limited response to immunotherapy observed in many STS [52].

Then, chemobiogram analysis on RMS1 patient-derived cells cultured both in standard monolayer culture and within 3D collagen-based scaffold showed the sensitivity of primary cells to some drugs currently used or under clinical evaluation for STS. In particular, RMS primary cells exposed to anthracycline-based regimens, the upfront treatment option in STS management including RMS, exhibited a high cytotoxic activity in both cell culture systems (Figure 7). This is in contrast with the poor response of the patient to doxorubicin therapy (Appendix A); however, this discrepancy could be explained by the gap that still exists between preclinical and clinical outcome. Indeed, patient-derived cell cultures could not completely recapitulate the tumor lesion microenvironment and behavior. This is in part due to the fact that primary cultures are not immortalized cells and, therefore, it is not possible to perform sequential treatments in order to assess the possible occurrence of chemoresistance [53]. Moreover, these results are consistent with daily clinical practice, confirming anthracycline-based regimens as a valuable option for the treatment of RMS. Moreover, the comparable cell viability among DOXO, DOXO/DACA and DOXO/CIS treatment groups in 2D provide evidences of the anthracycline efficacy in monoregimen. The same was observed in 3D culture system, in which DOXO efficacy was higher compared to DOXO/DACA and not significantly different compared to DOXO/CIS. The lower sensitivity of RMS cells to chemotherapy observed in 3D is explainable by the lower oxygen tension and molecules penetration within the collagen scaffold matrix. In this regard, we already showed that our 3D culture model enables the modeling of the tumor hypoxic state and the associated emergence of chemoresistance mechanisms [54]. The latter data is also observed with ETO and LENVA treatments group. In particular, ETO treatment group exhibited 25% cell death in 2D while it was not effective in 3D. LENVA treatment group exhibited 61% cell death in 2D while it was not effective in 3D. These data could also be linked to the mechanism of action of these drugs. Although we already demonstrated that our model is able to increase the expression of some STS associated-markers and to select a higher tumor cells percentage compared to standard 2D cultures [55], we can speculate that the lower sensitivity of 3D cultured RMS cells could be linked to a decreased expression of the target of these drugs in 3D. This could be especially true for a multikinase inhibitor such as lenvatinib, which acts via the inhibition of multiple targets interfering with the vascular endothelial formation. We can hypothesize that a more aggressive microenvironment such as that of a 3D culture system compared to 2D is responsible for the decreased survival of some of the patient-derived endothelial cells, which could be involved in the activity of the lenvatinib drug. This could explain the decrease in sensitivity observed in 3D, however, further research is needed to better elucidate the mechanism of action of these drugs.

As previously shown (Figure 4), upregulation of chemoresistance-related genes was detected in RMS1. In particular, *LAPTM4b*, a gene which seems to be involved in membrane trafficking of drugs such as anthracyclines, was upregulated while *LAPTM4a* was not. Although chemobiogram analysis (Figure 7) showed sensitivity to chemotherapy in both 2D and 3D RMS cultures, the above result is consistent with the clinical outcome observed in the patient, which showed a progression of disease after four cycles during the first line treatment with doxorubicin (Appendix A). These data suggest *LAPTM4b* as a predictive response marker in this histotype. More analyses are needed in order to confirm this data. Moreover, an upregulation of other chemoresistance-related genes such as *TP53I3*, *RAB22a,* and *S100p* was also detected. This data could in part explain the poor outcome of this specific entity to current chemotherapy. This is especially evident if comparing adult RMS to those observed in child counterparts. The above data could provide the rationale for targeting chemoresistance-related genes for reverting the sensibility of adult RMS to chemotherapy.

Finally, in silico analysis provides evidence of the potential role of *FGFR4* as a promising marker of poor prognosis and therapeutic target for these lesions, and that these features are in common with pediatric counterparts.

The study presents several limitations. First, due to the rarity of the investigated disease, only three patients were involved in the study. Moreover, the limited reliability of the preclinical models in reproducing the full spectrum of all tumor features, together with the heterogeneity of primary culture tumor cells and the lack of in vivo studies represent some criticisms and could affect the results.

## 4. Materials and Methods

### 4.1. Ethical Statement and Case Series

The study involved three adult patients affected by RMS. The study protocol was approved by IRST-Area Vasta Romagna Ethics Committee, approval no. 4751, 31 July 2015. The study was conducted according to the Good Clinical Practice standard operating procedures and 1975 Helsinki declaration. All patients gave informed consent for participation in the research study.

### 4.2. Histological and Immunohistochemical Analyses

Cytomorphological features of tumor tissues were analyzed through hematoxylin and eosin (H&E) staining. Briefly, resected tumor tissue was paraffin embedded and sectioned into 5-µm-thick slices using a microtome and stained using standard techniques. Protein expression was assessed by immunohistochemical analysis as previously reported [56]. In brief, 5-μm-thick sections cut from tumor tissues paraffin-embedded were de-paraffinized with xylene for 1 h, then rehydrated and incubated with antigen retrieval solution in a water bath at 98.5 °C for 30 min. After cooling, the sections were incubated for 10 min with 3% hydrogen peroxide solution and washed twice with demineralized water. Next, they were incubated with a 3% bovine serum albumin solution in PBS for 20 min and then incubated with antibody (MYOD-1, MDM2, SMA, CD34, desmin, myogenin, Ki-67, p63, S100, STAT6, myoglobin, AE1-AE3 cytokeratin, synaptophysin, CD99, CD45, GFAP, and ALK) at room temperature for 1 h. Staining was revealed using the streptavidin-biotin-peroxidase complex (ABC) method. Cell nuclei were counterstained with hematoxylin (Sigma Aldrich, Saint Louis, MO, USA). Cells were considered positive in the presence of brown nuclear immunostaining.

### 4.3. Real-Time PCR Analysis

mRNA was extracted from FFPE specimens sections (tumor and normal tissue selected by a sarcoma pathologist, the latter used as study control) using TRIzol Reagent (Invitrogen, Carlsbad, CA, USA) following the manufacturer’s instructions. Five hundred nanograms of extracted RNA were reverse-transcribed using iScript cDNA Synthesis Kit (BioRad, Hercules, CA, USA). Gene expression analysis was then carried out by Real-Time PCR using a 7500 Real-Time PCR System (Applied Biosystems, Foster City, CA, USA). A total volume of 20 µL containing both Taqman Universal PCR Master Mix (Applied Biosystems) and 2 µL of cDNA was used for the amplification step. The following markers were analyzed: *VIM*, *CDH1*, *TGFb*, *SNAIL1*, *MMP2*, *MMP9,* and *SLUG*. *HPRT* was used as a reference gene. SYBR Select Master Mix (Applied Biosystems) with 2 µL of cDNA was used for *TP53I3, LAPTM4A*, *RAB22A*, *S100P*, *CD109*, and *LAPTM4B* analysis. *GAPDH* was used as a reference gene. Specific target gene assays are reported in Appendix A. Relative expression was calculated with the 2(-delta delta C(T)) method.

### 4.4. Isolation of Patient-Derived RMS Primary Cells

RMS patient-derived primary cells (RMS1 patients) were isolated and established according to the protocol previously reported [35,57]. Briefly, surgical-resected tumor tissue was washed in PBS, sectioned, and finely crumbled with surgical scalpels. The obtained tumor fragments were enzymatically digested with a PBS and solution of collagenase type I (Millipore Corporation, Billerica, MA, USA) supplemented with 1:1 Dulbecco’s-modified Eagle’s medium (DMEM, Invitrogen, Darmstadt, Germany) for 15 min at 37 °C under stirring conditions. The sample was then stored overnight under stirring condition at room temperature. The day after, enzymatic digestion was stopped by adding DMEM supplemented with 10% fetal bovine serum (Invitrogen), 1% penicillin/streptomycin, and 1% glutamine. The obtained cell suspension was filtered with a 100 μm sterile mesh filter (CellTrics, Partec, Münster, Germany) and the isolated tumor cells were counted and seeded in standard monolayer cultures or in collagen-based scaffolds. For the 2D culture model, a cell density of 80,000 per cm^2^ was used. For the 3D culture system, the seeding was obtained following the simple soaking of the cell suspension in the dry scaffolds as previously reported [35], with a cell density of 500,000 cells/mm^3^. RMS primary cultures were maintained in DMEM supplemented with 10% fetal bovine serum (Invitrogen), 1% penicillin/streptomycin, and 1% glutamine at 37 °C in a 5% CO_2_ atmosphere. The medium was replaced daily. All the experiments were conducted using low-passage and proliferating primary cultures.

### 4.5. Establishment of Sclerosing RMS Patient-Derived Primary Culture

Cytomorphological features of patient-derived primary cells were analyzed through hematoxylin and eosin (H&E) staining. For patient-derived monolayer primary cultures, 100,000 cells were cytospinned onto glass slides, fixed with acetone for 10 min and with chloroform for 5 min, and downstream analyses were performed according to the manufacturer’s instructions. For patient-derived 3D primary culture models, 500,000 cells were cultured for 7 days in collagen-based scaffold and then paraffin-embedded. Five micrometer-thick slices were obtained and stained following the same protocols used for monolayer culture. For MYOD-1 immunohistochemical analysis, the same protocol described for tumor tissues was performed.

### 4.6. Building of a Collagen-Based Scaffold 3D Culture Model

Tridimensional collagen-based scaffold culture systems were synthesized in our laboratory as previously reported [24,58,59]. All reagents were purchased from Sigma Aldrich (St. Louis, MO, USA). Briefly, a 1 wt% suspension of bovine-derived insoluble type I microfibrillar collagen isolated from the Achilles tendon was dispersed in a 0.05 M of acetic acid solution. Next, 1 M of sodium hydroxide solution was added to the mixture obtaining a homogeneous suspension of fine collagen co-precipitates. The material was then cross-linked with a 1 wt% 1,4-butanediol diglycidyl ether (BDDGE) solution for 24 h at 25 °C to stabilize the collagen matrix and to control porosity and tortuosity. The obtained suspension was mixed through a homogenizer (IKA T18 Basic ULTRA-TURRAX, IKA, Staufen im Breisgau, Germany) at 20,000 rpm for 30 min at 4 °C and then centrifuged to remove air bubbles. The mixed solution was then frozen for 24 h and freeze-dried for 24 h. To consolidate the matrix architecture and achieve optimal levels of pore interconnectivity, the freeze-drying process was carried out with a controlled freezing and heating ramp from 25 °C to −35 °C and from −35 °C to 25 °C under vacuum conditions, *p* = 0.20 mbar. Finally, the scaffolds were sterilized through immersion in ethanol 70% for 1 h and then washed with PBS three times before using for in vitro analysis.

### 4.7. Chemobiogram Analysis

The efficacy of chemotherapy was assessed by MTT reduction assay. Briefly, tumor cells were seeded in both 96-well plates and 3D collagen-based scaffold at a density of 80,000 cells/cm^2^. Cells were allowed to recover for 3 days before drug exposure. The regimens were selected according to peak plasma concentration of each drug obtained from pharmacokinetic clinical data. In particular, 4 µg/mL of doxorubicin (DOXO) (Accord Healthcare Italia Ltd., Milan, Italy), 4.1 µg/mL of cisplatin (CIS) (Accord Healthcare Italia Ltd., Milan, Italy), 8 µg/mL of dacarbazine (DACA) (Medac Pharma Ltd., Rome, Italy) [23,60,61], 54 µg/mL of etoposide (ETO) (Teva Pharmaceutical Industries Ltd., Petah Tiqwa, Israel), [62] and 0.6 µg/mL of lenvatinib (LENVA) (Eisai Ltd., Milan, Italy) [63]. Cells survival percentage was assessed after 72 h of drug exposure. The experiments were performed twice.

### 4.8. Next-Generation Sequencing (NGS)

DNA and RNA samples were extracted from FFPE sections using the MagMAX FFPE DNA/RNA Ultra Kit (Thermo Fisher Scientific, Waltham, MA, USA) as previously described [64]. DNA and RNA concentrations were determined by fluorometric quantitation using Qubit 2.0 Fluorometer with Qubit DNA dsDNA HS Assay Kit and Qubit RNA HS Assay Kit (Thermo Fisher Scientific, Waltham, MA, USA) as appropriate. Subsequently, an RNA sample was used to perform the complementary DNA (cDNA) synthesis with SuperScript™ VILO™ cDNA Synthesis Kit (Thermo Fisher Scientific, Waltham, MA, USA) using 10 ng of RNA as input. DNA and RNA library were prepared with the Ion Chef System (Thermo Fisher Scientific, Waltham, MA, USA) using the previously prepared cDNA and a total of 12 ng DNA as input with the Oncomine Focus Assay, Chef-Ready Library Kit (Thermo Fisher Scientific, Waltham, MA, USA). Also, template preparation was performed on the Ion Chef System (Thermo Fisher Scientific, Waltham, MA, USA) using the Ion 510™ & Ion 520™ & Ion 530™ Kit–Chef. Sequencing was performed using the Ion GeneStudio™ S5 Plus System (Thermo Fisher Scientific, Waltham, MA, USA) and analyzed by next generation sequencing using a 52 gene panel (AmpliSeq for Illumina Focus Panel, San Diego, CA, USA) on MiSeq platform (Illumina, San Diego, CA, USA).

Data analysis was carried out using Ion Torrent Suite™ Browser version 5.12, to assess chip loading density, median read length, number of mapped reads, and the sample’s quality. Unaligned binary files (uBAM) were uploaded in the Ion Reporter™ Software (IR) 5.10 (ThermoFisher Scientific, Waltham, MA, USA) to perform sequence alignment and detection of SNVs, CNVs, 5’-to-3’ imbalance, and specific gene fusions. Variants were identified applying a custom filter chain that included alterations with minimum VAF 5% and a cut off of 500× coverage was set. For fusions, the minimum number of reads to consider a positive detection was >20. The Coverage Analysis plugin was applied to all data and used to assess amplicon coverage for regions of interest.

### 4.9. Evaluation of Microsatellite Instability (MSI) Status

Microsatellite instability status was assessed on the same tumor DNA used for NGS analysis. Briefly, the EasyPGX ready MSI kit (Diatech Pharmacogenetics, Jesi, Italy), which simultaneously analyzed eight markers (BAT-25, BAT-26, NR-21, NR-22, NR-24, NR-27, CAT-25, and MONO-27) was used (see complete list of gene in Appendix A). The EasyPGX^®^ Analysis Software, which permits the detection of alterations in the MSI status, comparing the results to a control sample included in the run, was used for data analysis.

### 4.10. In Silico Analysis

cBioPortal for Cancer Genomics (https://www.cbioportal.org/, accessed on 14 July 2021) [65] was used for gene mutations in silico analysis of 43 rhabdomyosarcomas [25].

### 4.11. Statistical Analysis

Three independent replicates were performed for each experiment. Outputs are presented as mean ± standard deviation (SD), or mean ± standard error (SE), as stated. Differences between groups were assessed by a two-tailed Student’s *t*-test and accepted as significant at *p* < 0.05.

## 5. Conclusions

Taken together, these results provide new insight into this poorly understood disease in adults. In particular, molecular analysis suggested some EMT and chemoresistance-related genes as promising markers for adult RMS. Moreover, this study provides the first indication for some of the drugs currently used or under clinical evaluation for the management of adult RMS. In particular, anthracycline-based regimen seems to be a valuable option for the treatment of this histotype.

To the best of our knowledge, this is the first translational work on an adult RMS case series using 3D patient-derived primary culture in which the molecular and the chemosensitivity profiling has been investigated. This exploratory study could represent a starting point for further research aimed at better characterizing the biological behavior and improving the clinical management of this uncommon disease.

## Figures and Tables

**Figure 1 ijms-22-11564-f001:**
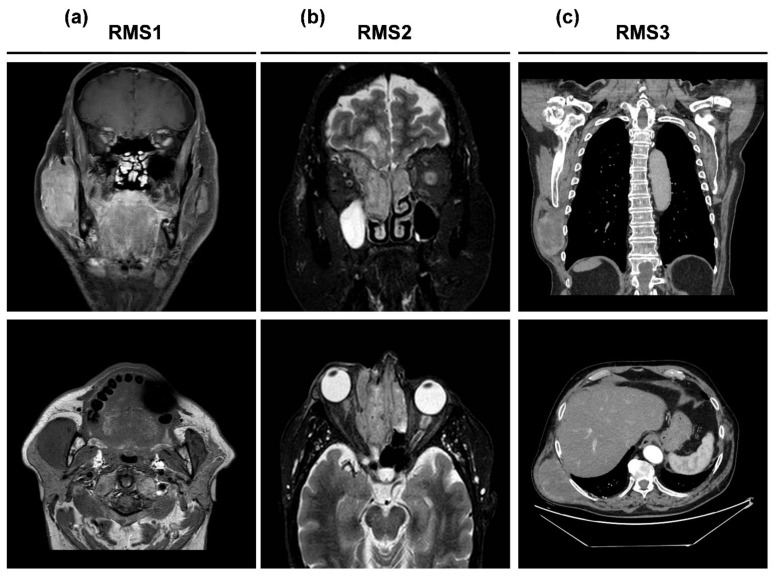
(**a**) Coronal and axial post-contrast MRI showing an heteroplastic expansive lesion involving a large part of the right masseter muscle measuring 33 mm × 36 mm (axial plane) and extension in the cranio-caudal direction of about 60 mm; (**b**) Coronal and axial post-contrast MRI showing an expansive ethmoidal formation, which also involves the cells of the ethmoidal labyrinth contralaterally. The lesion extends caudally to the right upper and middle nasal cavity, incorporating and displacing the middle turbinate inferiorly and cranially with invasion of the cribrosa lamina and the frontal plane, and in the right frontal-basal intracranial site, with an extension of about 8 mm. The tissue also infiltrates and invades the right orbital cavity in the supero-medial area with an extension of about 13 mm and a compressive-distorting effect on the eyeball, which appears to be displaced inferiorly laterally to the right and on the extrinsic musculature of the eye; (**c**) Coronal and axial post-contrast CT scan expansive lesion along the right posterior axillary line, below the great dorsal muscle, with axial diameters of 9 cm × 5 cm and cranio-caudal extension of approximately 9 cm.

**Figure 2 ijms-22-11564-f002:**
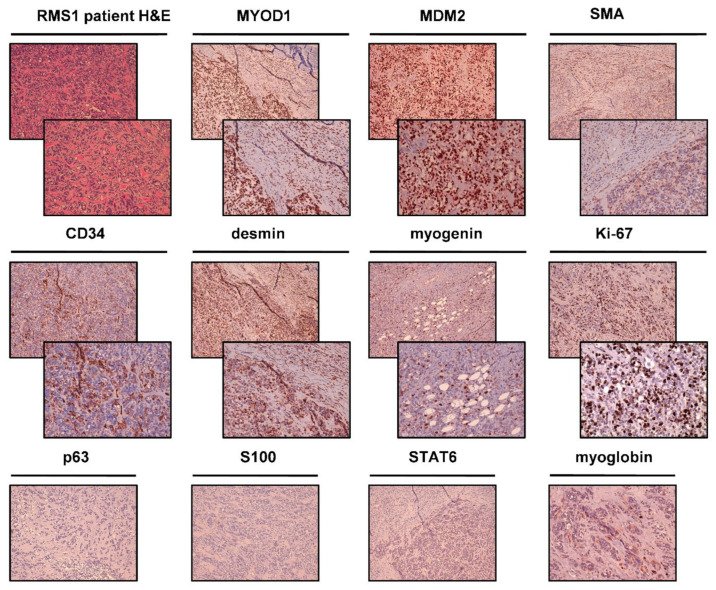
RMS1 patient histological and immunohistochemical analyses. H&E of the patient surgically resected tumor specimen. IHC analysis was performed on the patient surgically resected tumor specimen for the following markers: MYOD-1, MDM2, SMA, CD34, desmin, myogenin, Ki-67, p63, S100, STAT6, and myoglobin. For each marker, upper panels represent 10× magnification, and lower panels represent magnification 20× (negative markers are represented with only 10× magnification).

**Figure 3 ijms-22-11564-f003:**
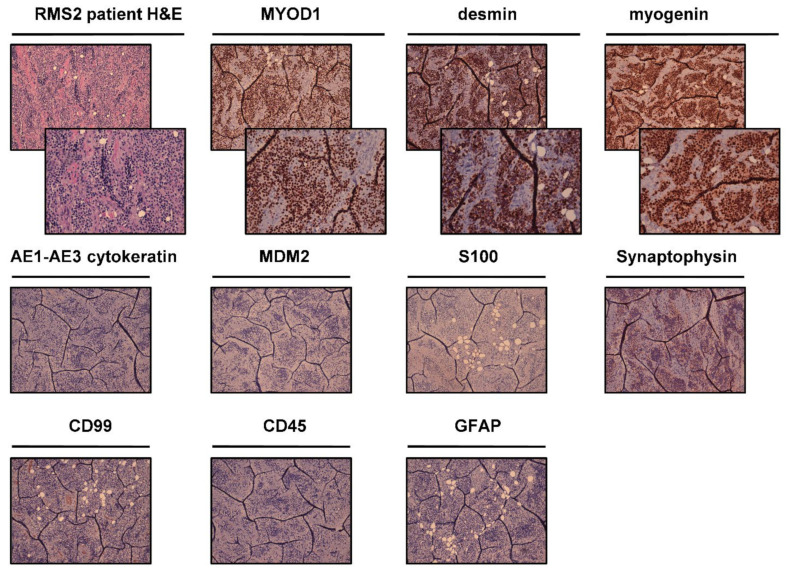
RMS2 patient histological and immunohistochemical analyses. H&E of the patient surgically resected tumor specimen. IHC analysis was performed on patient surgically resected tumor specimen for the following markers: MYOD-1, desmin, myogenin, AE1-AE4 cytokeratin, MDM2, S100, synaptophysin, CD99, CD45, and GFAP. For each marker, upper panels represent 10× magnification, and lower panels represent magnification 20× (negative markers are represented with only 10× magnification).

**Figure 4 ijms-22-11564-f004:**
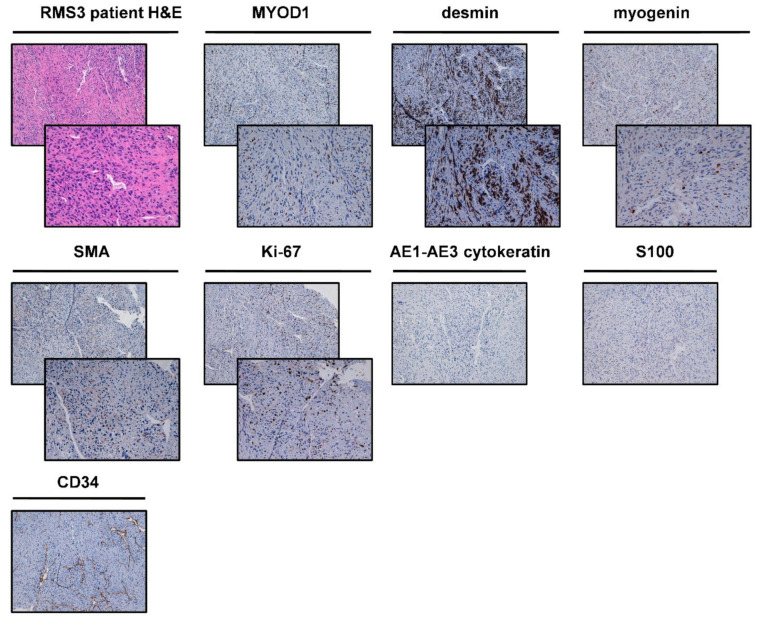
RMS3 patient histological and immunohistochemical analyses. H&E of the patient surgically resected tumor specimen. IHC analysis was performed on patient surgically resected tumor specimen for the following markers: MYOD-1, desmin, myogenin, SMA, CD34, AE1-AE4 cytokeratin, MDM2, S100, STAT6, and ALK. For each marker, upper panels represent 10× magnification, and lower panels represent magnification 20× (negative markers are represented with only 10× magnification).

**Figure 5 ijms-22-11564-f005:**
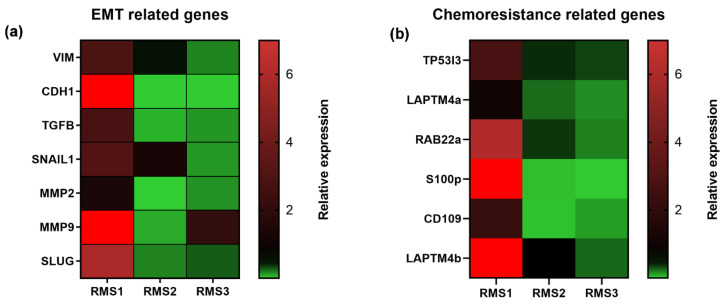
(**a**) Heat map comparison of the relative gene expression of selected EMT-associated genes in RMS tumor tissue compared to matched control tissue; (**b**) Heat map comparison of the relative gene expression of selected chemoresistance-associated genes in RMS tumor tissue compared to matched control tissue.

**Figure 6 ijms-22-11564-f006:**
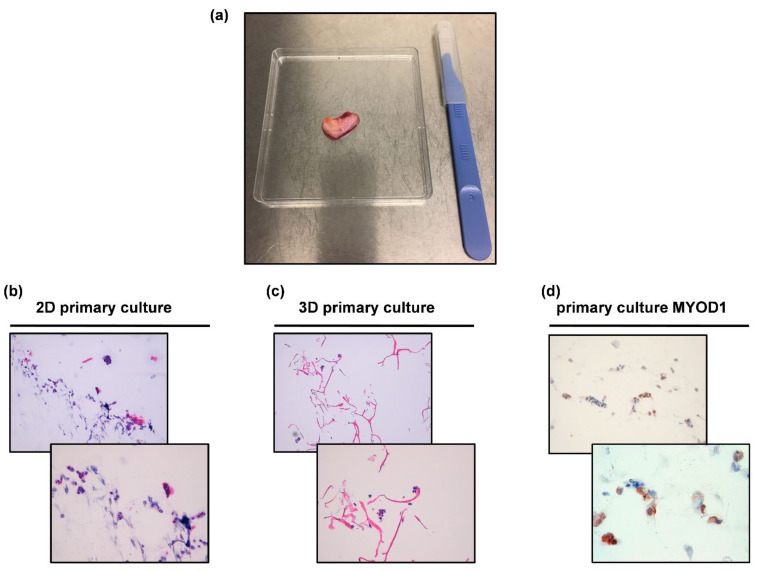
(**a**) Patient surgical specimen; (**b**) H&E of the 2D RMS patient-derived primary culture; (**c**) H&E of the 3D RMS patient-derived primary culture; (**d**) MYOD-1 staining by IHC in the RMS patient-derived primary cells (MYOD-1 positivity 70%). For each culture, upper panels represent 10× magnification, lower panels represent magnification 20×.

**Figure 7 ijms-22-11564-f007:**
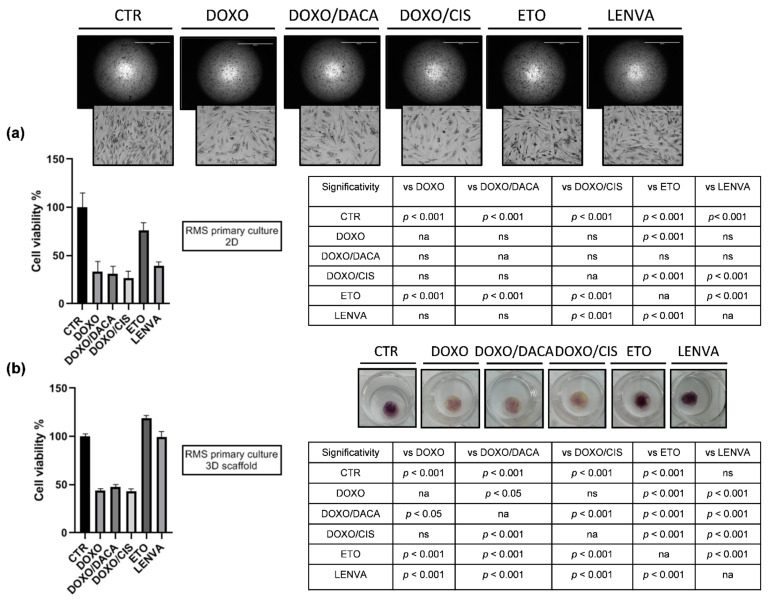
Chemobiogram analysis on (**a**) 2D (2× magnification, 2000 µm scale bar; 10× magnification, 400 µm scale bar) and (**b**) 3D patient-derived RMS primary culture. RMS primary cells were exposed to DOXO, DOXO/DACA, DOXO/CIS, ETO, and LENVA. Significant differences among treatments were accepted for *p* < 0.05.

**Table 1 ijms-22-11564-t001:** Clinicopathological characteristics of enrolled patients.

Patient	Gender	Age at Surgery	Site	Size (cm)	Histological Subtype	IHC Analysis	Surgical Margins	Radiotherapy Post-Surgery	Chemotherapy Post-Surgery	Follow-up Months
RMS1	male	74	masseterine parotid region	10 × 9 × 3	Spindle cell/sclerosing RMS	VIM+ MyoD1+ CD56+ Desmin+ Myoglobin+ Myogenin+ SMA+ CD34+ AE1-AE3 cytokeratin+ MDM2+ S100– STAT6– P63–	R1	adjuvant IMRT radiotherapy treatment 60 Gy	Doxorubicin4 cycles Gemcitabine 3 cycles	11
RMS2	female	72	ethmoid-orbital region	1.2 × 0.8	Alveolar RMS	MyoD1+ Desmin+ Myogenin+ AE1-AE3 cytokeratin– MDM2– S100– Synaptophysin– CD99– CD 45– GFAP–	na	na	na	7
RMS3	male	77	latissimus dorsi	15 × 11.5 × 7.5	Pleomorphic RMS	MyoD1+ Desmin+ Myogenin+ SMA+ CD34– AE1-AE3 cytokeratin– S100–	R0	na	na	1

**Table 2 ijms-22-11564-t002:** Translational comparison between the clinical outcome and in vitro analysis of enrolled patients.

Patient	Response to Chemotherapy	Survival Data	In Vitro Chemosensitivity (Cell Survival %)
RMS1	PD after Doxorubicin 4 cycles SD after Gemcitabine 3 cycles	deceased after 11 months of follow up	(2D) 33% Doxorubicin 31% Doxorubicin and Dacarbazine combination 26% Doxorubicin and Cisplatin combination 75% Etoposide 39% Lenvatinib (3D) 43% Doxorubicin 47% Doxorubicin and Dacarbazine combination 43% Doxorubicin and Cisplatin combination 100% Etoposide 99% Lenvatinib
RMS2	na	alive after 7 months of follow up	na
RMS3	na	alive after 1 month of follow up	na

PD, progression disease; SD, stable disease; na, not applicable.

## Data Availability

The datasets generated and/or analyzed during the current study are available from the corresponding author on reasonable request.

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
