# Peer review of "Deciphering the Genomic Landscape and Pharmacological Profile of Uncommon Entities of Adult Rhabdomyosarcomas"

_ijms, 2021, doi:10.3390/ijms222111564_

Round 1
Reviewer 1 Report
The paper by Alessandro De Vita, et al. entitled “Deciphering the genomic landscape and pharmacological profile of uncommon entities of adult rhabdomyosarcomas” is a research on the biological behavior of adult RMS cases. Pharmacological profiling showed the highest sensitivity with anthracycline-based regimen in both 2D and 3D culture systems. NGS analysis detected RAB3IP-HMGA2 in frame gene rearrangement and FGFR4 mutation. In silico analysis confirmed the mutation of FGFR4 as a promising marker for poor prognosis and a potential therapeutic target. The authors concluded that this exploratory study could represent a starting point for further research to improve the clinical management of adult RMS. This study is well-conducted and the results are presented in a scientifically sound manner. These findings will be of interest to readers, however, I have the following concerns on the current form.
Comments
1. The present manuscript lacks in vivo study, which is required to enhance the credibility of the results. The authors are encouraged to add in vivo study to validate their results if available. If not, this limitation should be mentioned in Discussion.
2. A translational comparison between the clinical outcome and the in vitro results is the main topic of this paper. A table summarizing response to chemotherapy, survival data, and chemosensitivity should be added.
Author Response
- The present manuscript lacks in vivo study, which is required to enhance the credibility of the results. The authors are encouraged to add in vivo study to validate their results if available. If not, this limitation should be mentioned in Discussion.
According to the reviewer's suggestion we have highlited the anavilability of in vivo study in the study limitation.
-A translational comparison between the clinical outcome and the in vitro results is the main topic of this paper. A table summarizing response to chemotherapy, survival data, and chemosensitivity should be added.
We thank the reviewer for pointing this out. As requested a table summary of the response to chemotherapy, survival data, and chemosensitivity have been included at the end of the result section.
Morevoer english language and style have been carefully revised.

Reviewer 2 Report
The paper is well-written and reports a detailed view of three cases of adult rhabdomyosarcomas. Given the rare disease and the genomic insight provided, it will be interesting for the clinicians and molecular pathologists involved in sarcoma management.
Author Response
The paper is well-written and reports a detailed view of three cases of adult rhabdomyosarcomas. Given the rare disease and the genomic insight provided, it will be interesting for the clinicians and molecular pathologists involved in sarcoma management.
Thank you for your interest in our work and for the time and effort.
English language and style have been carefully revised.
